# Multimodal Graph-based Variational Mixture of Experts Network for Zero-shot Multimodal Information Extraction

Submission Id: 308*

## Abstract

Multimodal information extraction on social media is a series of fundamental tasks to construct the multimodal knowledge graph. The tasks aim to extract the structural information in free texts with the incorporate images, including: multimodal named entity typing and multimodal relation extraction. However, the growing number of multimodal data implies a growing category set and the newly emerged entity types or relations should be recognized without additional training. To address the aforementioned challenges, we focus on the zero-shot multimodal information extraction task which requires to utilize textual and visual modalities for identifying previously unseen categories in a zero-shot manner. Compared with the text-based zero-shot information extraction models, the existing multimodal ones make the textual and visual modalities aligned directly and exploit various fusion strategies to improve their generalization ability. But the existing methods only align the global representations of multimodal data and ignore the fine-grained semantic correlation of the text-image pairs and samples. Therefore, we propose the multimodal graph-based variational mixture of experts network (MG-VMoE) which takes the MoE network as the backbone and exploits the sparse expert weights for aligning the multimodal representations in a fine-grained way. Considering to learn the informative and aligned representations of multimodal data, we design each expert network as a variational information bottleneck to process the two modalities in a uni-backbone. Moreover, we do not only model the correlation of the text-image pair inner a sample, but also propose the multimodal graph-based virtual adversarial training to learn the semantic correlation between the samples. The experimental results on the two benchmark datasets demonstrate the superiority of MG-VMoE over the baselines.

## CCS Concepts

• **Information systems → Multimedia and multimodal retrieval**; • **Computing methodologies → Information extraction**.

## Keywords

multimodal information extraction, zero-shot learning, multimodal representation learning

**ACM Reference Format:**
Anonymous Author(s). 2018. Multimodal Graph-based Variational Mixture of Experts Network for Zero-shot Multimodal Information Extraction. In *Proceedings of Make sure to enter the correct conference title from your rights confirmation emai (Conference acronym 'XX)*. ACM, New York, NY, USA, 9 pages. https://doi.org/XXXXXXX.XXXXXXX

## 1 Introduction

Extracting structural information from free text in conjunction with images, in order to construct a multimodal knowledge graph [26, 31], constitutes a series of fundamental tasks known as multimodal information extraction (MIE). The MIE tasks are associated with the entity information to complete the specific tasks including: multimodal named entity typing (MET) [24] and multimodal relation extraction (MRE) [27]. Compared with the text-based IE models, the multimodal-based ones are proposed to capture the correlations of textual and visual contents with various fusion strategies for effective entity and relation classification [29].

In practical situations, however, the number of intricate entity types and relations is continually expanding, necessitating additional human input for annotating every novel category that arises. To address the above issue, the introduction of zero-shot learning into text-based information extraction (ZS-IE) models facilitated the identification of novel categories of entity types or relations without requiring additional training. The existing ZS-IE approaches primarily concentrate on the textual modality, leveraging pre-trained language models such as BERT to extract entity features for constructing representations of type or relation prototypes. Ma et al. [11] and Chen and Li [2] respectively considered the names of types and relations as the prototypical knowledge for recognizing the novel categories. Moreover, the attention mechanism is applied on the ZS-IE models to extract the fine-grained context representations implied in the external descriptions [16, 25]. Despite of the descriptions of categories, the multi-source knowledge of them were exploited to enhance the ZS-IE models in the fusion [3] or augmentation [7] way. With the development of social media, the growing number of multimodal data implies expanding category set of types and relations. And the above methods are focused on the textual modality while ignoring the visual modality.

The vital challenge to exploit multimodal data is to bridge the semantic gap between the two modalities. The previous MIE models proposed the different fusion strategies or alignment modules to extract the useful multimodal representations. Zheng et al. [27] designed the dual graph-based multimdoal alignment and fusion modules to improve the MRE performance. Zhang et al. [24] exploited the cross-modal transformer to obtain the multimodal representations for modeling the MET task. However, the current MIE models lack efficiency in recognizing newly emerged entity types or relationships on social media without additional training. This is due to the challenge posed by the diversity of textual and visual contents,

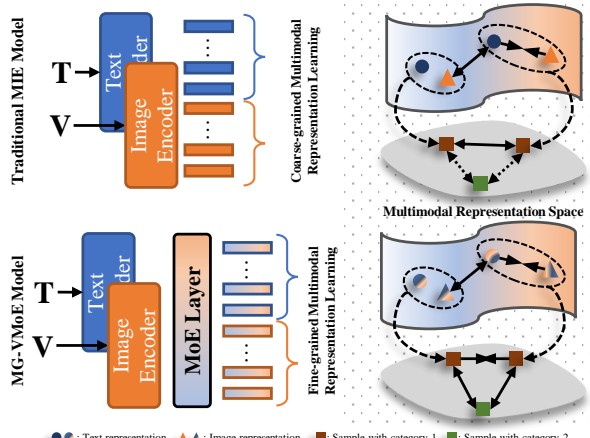

**Figure 1: The multimodal representation space comparison between the traditional MIE model and the MG-VMoE model.**

which span a wide range of entities and for which current models are unable to effectively bridge the semantic gap between the two modalities. As illustrated in Figure 1, traditional MIE models rely on coarse-grained multimodal representation learning to align the global features of text-image pairs within samples. Since samples within the same category can exhibit significant semantic variation in their texts and images, this approach is insufficient for capturing fine-grained semantic correlations between the two modalities at the token level and for clustering multimodal samples of the same category which constraints its ability to establish connections between multimodal samples and prototypical categories.

To address the above mentioned limitations, we propose the **m**ultimodal **g**raph-based **v**ariational **m**ixture **of** **e**xperts (MG-VMoE) network to tackle the zero-shot MIE (ZS-MIE) task. The MG-VMoE network is based on the fine-grained multimodal representation learning which consists of the architectures and the specific training method. For capturing semantic correlations between the two modalities at the token level, we utilize the mixture of experts (MoE) network as the backbone which exploits the sparse expert weights to align the textual and visual token representations in a fine-grained way. With the purpose to model the informative and aligned representations of multimodal data, we design each expert network as a variational information bottleneck (VIB) to handle the two modalities in a uni-backbone. For clustering samples belonging to the same category, we propose a multimodal graph-based virtual adversarial training method to capture the semantic correlations between multimodal samples. Ultimately, we fuse textual entity representations with multimodal ones through an attention layer and measure the semantic similarity between the fusion features and prototypical ones of different categories for recognition. The contributions of this manuscript can be summarized as follows:

- We present the zero-shot multimodal information extraction (ZS-MIE) task which leverages the text and image pairs to extract the novel knowledge such as: entity types or relations on social media without additional training.
- We propose a multimodal graph-based variational mixture of experts (MG-VMoE) network based on the fine-grained multimodal representation learning. Not only does the network utilize the VMoE architecture to model the aligned

multimodal representations within individual samples, but it also leverages multimodal graph-based virtual adversarial training to capture the semantic correlations existing between different samples.
- We conduct the extensive experiments on the two benchmark MIE datasets and the experimental results demonstrate the superiority of the proposed model over baselines.

## 2 Related Work

### 2.1 Zero-shot Information Extraction

Information extraction (IE) encompasses a range of tasks, notably named entity typing and relation extraction, aimed at distilling structural information from unstructured texts for the purpose of constructing comprehensive knowledge graphs [28]. Considering to recognize the unseen categories like: entity types or relations without additional training, the zero-shot learning was introduced into traditional information extraction (ZS-IE) tasks. The vital challenge for ZS-IE is to learn generalizable representations of entities and prototypical knowledge of categories.

For zero-shot named entity typing (ZS-ET), Ma et al. [11] firstly proposed a label embedding method to encode the prototypical knowledge of types with textual embeddings, and bridge the semantic correlation between entity mentions and types. Ren et al. [17] employed the attention mechanism to extract local features that are relevant to the types, with a focus on the nuanced semantic representations of both mentions and their contexts. Zhang et al. [23] devised the ZS-ET model, augmented with memory capabilities, to retain observed types as memory elements and facilitate knowledge transfer from known to unknown types, thereby explicitly capturing the semantic relationship between them. Furthermore, auxiliary data including descriptions sourced from websites was integrated into the ZS-ET model to augment the representation of mentions and types [3, 16]. For zero-shot relation extraction (ZS-RE), Chen and Li [2] initially leveraged BERT to acquire two functions, which project entities and relation descriptions into an embedding space by concurrently minimizing the distances between them and subsequently categorizing their corresponding relations. Zhao et al. [25] introduced a fine-grained semantic matching method, which dissects the overall sentence-level similarity score into distinct components for entity and context matching. Gong and Eldardiry [7] presented a prompt-driven model that augments semantic knowledge by creating instances featuring unseen relations from existing instances with known relations.

In essence, the current ZS-IE approaches solely concentrate on textual modality, neglecting the potential semantic enrichment from visual content that could strengthen entity representations. In contrast to these endeavors, our focus lies in the ZS-MIE task, aimed at extracting innovative structural knowledge embedded within multimodal data sourced from social media platforms.

### 2.2 Mulitmodal Information Extraction

As the volume of multimodal data continues to expand, researchers have recognized the need to capture the intricate semantic information within these data. Consequently, they have extended the traditional IE tasks to encompass multimodal IE, resulting in improved outcomes. Moon et al. [14] initially broadened the scope of

traditional text-based named entity recognition to encompass multimodal named entity recognition (MER), subsequently introducing the modality attention module concept as a means to integrate textual information with image data, thereby enhancing the accuracy of sequence label predictions. Zhang et al. [24] proposed to incorporate visual objects and exploit the cross-modal transformer to obtain multimodal representations for tackling the multimodal named entity typing (MET) task firstly. Zheng et al. [27] introduced the multimodal relation extraction (MRE) task, leveraging visual modality to bolster the semantic representations of textual modality. Cui et al. [4] exploited the variational information bottleneck to extract effective multimodal representations for the MIE tasks.

In summary, the aforementioned multimodal information extraction tasks operate within a supervised framework, aiming to establish a mapping from multimodal data to predefined labels. However, our focus lies in zero-shot multimodal information extraction (ZS-MIE), which endeavors to recognize unseen categories without requiring additional training. In contrast to supervised learning that relies on abundant labeled data, zero-shot learning prioritizes the development of generalizable representations for both samples and semantic labels, enabling the inference of samples belonging to unobserved categories.

## 3 Preliminary

Before introducing the details of the proposed model, we formalize the problem of zero-shot multimodal information extraction (ZS-MIE). We obtain the training dataset $\mathcal{D}_{train}$ with the seen category set $\mathcal{Y}_s$ and the test dataset $\mathcal{D}_{test}$ with the unseen category set $\mathcal{Y}_u$. The seen category set is defined as $\mathcal{Y}_s = \{y_1^s, y_2^s, \ldots, y_{|\mathcal{Y}_s|}^s\}$ and the unseen one is denoted as $\mathcal{Y}_u = \{y_1^u, y_2^u, \ldots, y_{|\mathcal{Y}_u|}^u\}$. And the two aforementioned category sets are mutually disjoint. Each sample of the datasets is denoted as a tuple $S = (T, V, E, Y)$ where $T$ denotes a natural language sentence, $V$ is the image coupled with the sentence, $E$ represents the entity information such as: the location in the sentence, and $Y$ is the label for specific tasks.

According to zero-shot learning, we need to train a ZS-MIE model $\mathcal{M}$, i.e., $\mathcal{M}(S) \rightarrow Y \in \mathcal{Y}_s$ based on the training set $\mathcal{D}_{train}$. The ZS-MIE model can be defined as $\mathcal{M} : \mathcal{F} \times \mathcal{G}$ where learnable functions $\mathcal{F}$ and $\mathcal{G}$ project the multimodal data and category semantic information into the shared embedding space respectively. During the training procedure, we optimize $\mathcal{M}$ by minimizing the distance between the multimodal representation $\mathcal{F}(T, V, E)$ and the category semantic representation $\mathcal{G}(Y)$. To recognize the unseen categories, we employ the acquired functions $\mathcal{F}$ and $\mathcal{G}$ to encode the multimodal representation of a sample from $\mathcal{D}_{test}$ as well as the semantic representations of the unseen categories within $\mathcal{Y}_u$, and then pinpoint the nearest unseen category as the prediction.

## 4 Methodology

In this section, we introduce the multimodal graph-based variational mixture of experts (MG-VMoE) network for zero-shot multimodal information extraction as shown in Figure 2. The overall framework is based on the fine-grained multimodal representation learning which consists of the architectures and the specific training method. In order to build up the model, the details of our model can be summarized as the following parts: (1) Firstly, we extract the multimodal input representations of the samples with the pre-trained language and vision models. (2) Secondly, to capture the fine-grained semantic correlation between the text and image token representations, we propose the VMoE network which models the informative and aligned representations of multimodal data in a uniform backbone. (3) Thirdly, we propose the multimodal-graph based virtual adversarial training to model the semantic correlation between multimdoal samples and keep samples belonging to the same category more clustered. (4) Eventually, to identify unseen categories of samples, we calculate the distances between the multimodal features of samples and the label semantic ones.

### 4.1 Multimodal Input Representation

Given the multimodal samples which consists of texts and images, we need to map them into the dense representations for deep neural networks. For the images, we make use of ViT [6] to extract the visual representations of images. In contrast to pre-trained models grounded in convolutional neural networks, such as ResNet, ViT divides the image into patches and employs transformer modules to preserve local visual information within its high-level representation framework. We input the image $V$ of the sample into ViT and obtain the visual representations which are denoted as $\mathbf{V} = \{\mathbf{v}_1, \mathbf{v}_2, \ldots, \mathbf{v}_{|V|}\}$ where $\mathbf{v}_i \in \mathbb{R}^d$ and $|V|$ is the number of feature vectors output from ViT.

As for the texts, we denote the original sentence with $|S|$ words as $S = \{w_1, w_2, \ldots, w_{|S|}\}$. To take advantage of pre-trained language models, we utilize BERT [5] as the text encoder to map discrete words into the dense representations. Before feeding the sentence into BERT, we should pre-process it with special tokens. Each sentence requires the insertion of the reserved tokens [CLS] at the beginning and [SEP] at the end. Furthermore, the entity information holds significant importance for ZS-MIE tasks, which encompass MET and MRE. The reserved tokens [E1], [/E1] (and [E2], [/E2]) are inserted into the sentence to mark the begin and end of the entities from the entity set $E$ [29]. Formally, the extended sentence with the special tokens is denoted as . And the calculation process of sentence representations can be simplified as $\mathbf{T} = BERT(\tilde{S})$ where $\mathbf{T} = \{\mathbf{t}_{[CLS]}, \mathbf{t}_1, \ldots, \mathbf{t}_{[SEP]}\} \in \mathbb{R}^{d \times |T|}$ and $|T|$ is the token number of the extended sentence. We extract the entity representation according to the specific tasks.

**Multimodal Named Entity Typing (MET).** To keep the entity and contextual information for MET task, we define the entity representation as: $\mathbf{E} = [\mathbf{t}_{[CLS]} \oplus \mathbf{t}_{[E1]}] \in \mathbb{R}^{2d}$ where $\mathbf{t}_{[CLS]}$ and $\mathbf{t}_{[E1]}$ represent the features of the tokens [CLS] and [E1] respectively, and $\oplus$ is the vector concatenation operation.

**Multimodal Relation Extraction (MRE).** Analogous to the MET task, we require extracting the entity representation by incorporating both the head and tail entities as: $\mathbf{E} = [\mathbf{t}_{[CLS]} \oplus \mathbf{t}_{[E1]} \oplus \mathbf{t}_{[E2]}] \in \mathbb{R}^{3d}$ where $\mathbf{t}_{[E1]}$ and $\mathbf{t}_{[E2]}$ denote the beginning tokens [E1] and [E2] of head and tail entities respectively.

Furthermore, we consider the semantic information of categories such as: the names of labels as the prototypical knowledge. Given the seen category set $\mathcal{Y}_s = \{y_1^s, y_2^s, \ldots, y_{|\mathcal{Y}_s|}^s\}$, we consider each category name as a sentence $\{w_1, w_2, \ldots, w_l\}$ and feed it into BERT to acquire the textual representations $\{\mathbf{r}_0, \mathbf{r}_1, \ldots, \mathbf{r}_{l+1}\}$. The semantic representation of the category is calculated as $\mathbf{R} = \frac{1}{l+2} \sum_{i=0}^{l+1} \mathbf{r}_i$.

Figure 2: The overall framework of the multimodal graph-based variational mixture of experts (MG-VMoE) network for zero-shot multimodal information extraction. The upper part is the samples of multimodal named entity typing and multimodal relation extraction. The lower left is the multimodal backbone network based on VMoE and the lower right is the multimodal graph-based vitrual adversarial training.

Therefore, the prototypical representations of the seen category set is defined as $\mathbf{C}_s = \{\mathbf{R}_1^s, \mathbf{R}_2^s, \ldots, \mathbf{R}_{|\mathcal{Y}_s|}^s\}$.

## 4.2 Variational Mixture of Experts Network

The multimodal representations from the pre-trained language and vision models exist in their respective modality spaces. To address the semantic discrepancy between the two modalities, the existing MIE models frequently resort to contrastive learning to harmonize the multimodal representations emanating from pre-trained language and vision models [22]. Nonetheless, the aforementioned coarse-grained approach to multimodal representation learning primarily emphasizes the holistic representations of multimodal data, neglecting to model the local semantic relationships between text and image token representations. Therefore, we propose the variational mixture of experts (VMoE) network as the backbone to make the multimodal data aligned uniformly. The traditional MoE network consists of the router module and the expert modules [15]. Upon inputting a feature vector into the MoE network, the router module initially determines which expert modules to activate, guided by the input data's characteristics. Subsequently, each activated expert module processes the input data individually, producing respective outputs. Ultimately, these outputs undergo a weighted summation process, with weights assigned by the router module, to yield the fused prediction result [1]. Compared with the traditional MoE network, we formalize each expert module as the variational information bottleneck (VIB) [4] which can keep the multimodal representations from VMoE informative and aligned.

Given the textual representation $\mathbf{T}$ and the visual one $\mathbf{V}$, we combine them with the direct concatenation as $\mathbf{M} = [\mathbf{V}; \mathbf{T}] \in \mathbb{R}^{d \times (|V|+|T|)}$. The expert module, for multimodal representation $\mathbf{M}$, is structured as VIB that learns a latent representation $\mathbf{Z}$ while preserving sufficient information from $\mathbf{M}$ crucial for prediction. The information bottleneck (IB) [18] is formalized as follows:

$$\mathcal{L}_{IB} = \beta \cdot I(\mathbf{M}, \mathbf{Z}) - I(\mathbf{Z}, Y) \qquad (1)$$

where $I(\cdot)$ is the mutual information (MI) between two variables. To reduce irrelevant information, we minimize the mutual information $I(\mathbf{M}, \mathbf{Z})$ between the input representation $\mathbf{M}$ and the latent representation $\mathbf{Z}$ generated by the expert module. Additionally, to ensure sufficient information for prediction, we maximize the mutual information $I(\mathbf{Z}, Y)$ between the latent representation $\mathbf{Z}$ and the target label $Y$. Considering that the MI is computationally intractable for deep neural networks [4], we utilize the variational manner to encode the latent representation. Therefore, the latent gaussian distributional variable $\mathbf{Z}$ is defined as follows:

$$\mathbf{Z} \sim \mathcal{N}(\boldsymbol{\mu}, \boldsymbol{\sigma}^2), \quad \boldsymbol{\mu} = FFNN(\mathbf{M}; \theta_\mu), \quad \boldsymbol{\sigma} = FFNN(\mathbf{M}; \theta_\sigma) \quad (2)$$

where $\boldsymbol{\mu}$ and $\boldsymbol{\sigma}$ are the mean and standard deviation vectors, and $FFNN(\cdot; \theta)$ is short for the feed-forward neural network with the trainable parameter $\theta$. We use the re-parameterization trick [9] to perform the equivalent sampling to obtain the latent representation $\mathbf{Z}$ as the following equation:

$$\mathbf{Z} = \boldsymbol{\mu} + \boldsymbol{\sigma} \odot \epsilon, \quad \epsilon \sim \mathcal{N}(0, \mathbf{I}) \qquad (3)$$

where $\odot$ is the element-wise production and $\mathbf{Z} \in \mathbb{R}^{d \times (|V|+|T|)}$ is the representation from each expert module. And for the $K$ expert modules, the combination of their output latent representations is denoted as: $\{\mathbf{Z}_i | \mathbf{Z}_i \sim \mathcal{N}(\boldsymbol{\mu}_i, \boldsymbol{\sigma}_i^2), i = 1, 2, \ldots, K\}$. To fuse the representations from different expert modules, the router module predicts the gating weights corresponding to the $K$ expert modules. The router module is implemented with the dense connection layer, and the gating weights are calculated as $\mathbf{G} = softmax(\mathbf{W}_g^T \mathbf{M} + \mathbf{b}_g) \in \mathbb{R}^{K \times (|V|+|T|)}$ where $\mathbf{W}_g \in \mathbb{R}^{d \times K}$ and $\mathbf{b}_g \in \mathbb{R}^K$ are the trainable parameters. The fusion multimodal representations are calculated as: $\mathbf{H} = \sum_{i=1}^K \mathbf{Z}_i \cdot \mathbf{G}_i$ and the fusion textual and visual token features are $\tilde{\mathbf{V}} = \{\mathbf{h}_i\}_{i=1}^{|V|}$ and $\tilde{\mathbf{T}} = \{\mathbf{h}_i\}_{i=|V|+1}^{|V|+|T|}$. Considering to sparsely activate expert modules for each token representation, we exploit the entropy auxiliary loss to optimize the router module

as the following equation:

$$\mathcal{L}_{aux} = -\frac{1}{|V|+|T|} \sum_{j=1}^{|V|+|T|} \sum_{i=1}^{K} G_{i,j} \log G_{i,j} \quad (4)$$

where $G_{i,j}$ is the weight score of the $i$-th expert module to the $j$-th token representation.

To optimize the model by the IB principle, we can estimate $I(\mathbf{M}, \mathbf{Z})$ as follows:

$$I(\mathbf{M}, \mathbf{Z}) = KL(p(\mathbf{Z}|\mathbf{M}) \parallel p(\mathbf{Z})) \le KL(q(\mathbf{Z}|\mathbf{M}) \parallel p(\mathbf{Z})) \quad (5)$$

where the posterior distribution $p(\mathbf{Z}|\mathbf{M})$ could be approximated by the variational posterior distribution $q(\mathbf{Z}|\mathbf{M})$ and the prior distribution $p(\mathbf{Z})$ is assumed as normal Gaussian distribution. Therefore, the regularization loss of VIB for the latent representations from $K$ expert modules is defined as follows:

$$\mathcal{L}_{reg} = \sum_{i=1}^{K} KL(q(\mathbf{Z}_i|\mathbf{M}) \parallel p(\mathbf{Z}_i)) = \sum_{i=1}^{K} KL(\mathcal{N}(\boldsymbol{\mu}_i, \boldsymbol{\sigma}_i^2) \parallel \mathcal{N}(0, \mathbf{I})) \quad (6)$$

To keep the latent representations related to task labels, we maximize $I(\mathbf{Z}, Y)$ with the variational lower bound [4] of it as follows:

$$I(\mathbf{Z}, Y) = \mathbb{E}_{p(\mathbf{Z}|\mathbf{M})} \left[ \log p(Y|\mathbf{M}) \right] \ge \mathbb{E}_{q(\mathbf{Z}|\mathbf{M})} \left[ \log p(Y|\mathbf{M}) \right]. \quad (7)$$

The above equation could be optimized by the task loss function such as ranking loss function for ZS-MIE tasks.

## 4.3 Multimodal Graph-based Virtual Adversarial Training

To bridge the semantic gap between the textual and visual modalities, the multimodal methods [1, 22] always utilize the contrastive learning to align the text and image representations of a sample. Given a batch of $N$ samples $\{(T_i, V_i)\}_{i=1}^{N}$, we feed them into the multimodal representation encoder which includes the pre-trained language and vision models stacked with VMoE network to obtain the fusion multimodal representations as $\{\mathbf{H}_i | \mathbf{H}_i = [\tilde{\mathbf{V}}_i; \tilde{\mathbf{T}}_i], i = 1, 2, \ldots, N\}$. We average the fusion textual and visual token features as the global representations $\{(\bar{\mathbf{T}}_i, \bar{\mathbf{V}}_i) | \bar{\mathbf{T}}_i = \frac{1}{|T|} \sum_{j=1}^{|T|} \tilde{\mathbf{t}}_{i,j}, \bar{\mathbf{V}}_i = \frac{1}{|V|} \sum_{j=1}^{|V|} \tilde{\mathbf{v}}_{i,j}, i = 1, 2, \ldots, N\}$. The objective of multimodal contrastive learning is to discern matched pairs from among $N \times N$ potential image-text combinations, ensuring that representations of paired inputs are positioned closer together in the representation space compared to those of unpaired inputs. Therefore, the multimodal contrastive learning for one batch is defined as follows:

$$\mathcal{L}_{cl} = \sum_{i=1}^{N} -\frac{1}{2}(\log \frac{\exp(\bar{\mathbf{T}}_i^T \bar{\mathbf{V}}_i)}{\sum_{j=1}^{N} \exp(\bar{\mathbf{T}}_i^T \bar{\mathbf{V}}_j)} + \log \frac{\exp(\bar{\mathbf{T}}_i^T \bar{\mathbf{V}}_i)}{\sum_{j=1}^{N} \exp(\bar{\mathbf{T}}_j^T \bar{\mathbf{V}}_i)}) \quad (8)$$

However, the aforementioned coarse-grained contrastive learning approach solely emphasizes the semantic coherence between text-image pairs within individual samples, falling short in clustering samples of the same category while simultaneously discerning intricate semantic nuances within multimodal data. This limitation ultimately restricts its capacity to forge fine-grained correlation between multimodal samples and prototypical knowledge. Therefore, we propose the multimodal-graph based virtual adversarial training (MG-VAT) to model the semantic correlation

between samples. Given a batch of multimodal samples, we integrate the global textual and visual representations as a whole $\mathbf{P} = \{\bar{\mathbf{H}}_i | \bar{\mathbf{H}}_i = [\bar{\mathbf{V}}_i \oplus \bar{\mathbf{T}}_i], i = 1, 2, \ldots, N\} \in \mathbb{R}^{2d \times N}$. To measure the semantic similarities of samples, we construct the multimodal sample correlation graph $\mathbf{A} = \{a_{i,j} | i, j \in \{1, 2, \ldots, N\}\} \in \mathbb{R}^{N \times N}$. And the elements in the graph are calculated as follows:

$$a_{i,j} = a_{j,i} = (1 + \frac{\bar{\mathbf{H}}_i^T \bar{\mathbf{H}}_j}{\|\bar{\mathbf{H}}_i\|_2 \|\bar{\mathbf{H}}_i\|_2})/2. \quad (9)$$

We utilize the graph information to aggregate the multimodal representations of relevant samples as: $\hat{\mathbf{P}} = \{\hat{\mathbf{H}}_i\}_{i=1}^{N} = \mathbf{PA}$ and the irrelevant ones as: $\hat{\mathbf{P}}' = \{\hat{\mathbf{H}}_i'\}_{i=1}^{N} = \mathbf{P}(\mathbf{1} - \mathbf{A})$. For each sample, we can measure its relative semantic correlation score with other relevant and irrelevant samples in the batch as:

$$\mathbf{s}_i = softmax([\bar{\mathbf{H}}_i^T \hat{\mathbf{H}}_i; \bar{\mathbf{H}}_i^T \hat{\mathbf{H}}_i']) \in \mathbb{R}^2 \quad (10)$$

and the score of the batch is denoted as $\mathbf{S} = \{\mathbf{s}_i\}_{i=1}^{N} \in \mathbb{R}^{2 \times N}$.

To enhance the robustness of multimodal representations of samples, we employ VAT [8, 13] to minimize the KL divergence between the relative semantic correlation scores of original samples and those of adversarial samples based on the multimodal sample correlation graph. We introduce the perturbation vector $\boldsymbol{\tau} \in \mathbb{R}^{2d}$ to generate the multimodal representations of adversarial samples as $\tilde{\mathbf{P}} = \{\tilde{\mathbf{H}}_i | \tilde{\mathbf{H}}_i = \bar{\mathbf{H}}_i + \boldsymbol{\tau}, i = 1, 2, \ldots, N\}$. Based on the given graph $\mathbf{A}$, we can also calculate the relative semantic correlation scores of adversarial samples as $\tilde{\mathbf{S}}$. In order to reduce the influence of perturbation, the adversarial loss for VAT is defined as follows:

$$\mathcal{L}_{vat} = KL(p(\mathbf{S}|\mathbf{P}) \parallel p(\tilde{\mathbf{S}}|\tilde{\mathbf{P}})). \quad (11)$$

Moreover, to compute the worst perturbation which can significantly improve multimodal representations, we can optimize the perturbation $\boldsymbol{\tau}$ by the following objective function:

$$\arg\max_{\boldsymbol{\tau}} KL(p(\mathbf{S}|\mathbf{P}) \parallel p(\tilde{\mathbf{S}}|\tilde{\mathbf{P}})) - \|\boldsymbol{\tau}\|_2. \quad (12)$$

## 4.4 Training and Inference Procedure

Given the fusion multimodal representations $\mathbf{H}$ and the textual entity representation $\mathbf{E}$, we utilize the attention mechanism [30] to extract the local features of the former which are related to the latter. The attention score is defined as $\alpha_i = \frac{\exp(\mathbf{W}_a^T [\mathbf{h}_i \oplus \mathbf{E}] + \mathbf{b}_a)}{\sum_{j=1}^{|V|+|T|} \exp(\mathbf{W}_a^T [\mathbf{h}_j \oplus \mathbf{E}] + \mathbf{b}_a)}$ where $\mathbf{W}_a \in \mathbb{R}^{d+|\mathbf{E}|}$ and $\mathbf{b}_a \in \mathbb{R}$ are the trainable parameters and $|\mathbf{E}|$ is the dimension number of textual entity representation. Therefore, the entity-aware multimodal representation is calculated as $\mathbf{U} = \sum_{i=1}^{|V|+|T|} \alpha_i \mathbf{h}_i$. For predicting the category, we concatenate the textual entity representation and entity-aware one as $\tilde{\mathbf{U}} = [\mathbf{U} \oplus \mathbf{E}]$ and regard the the semantic representations $\mathbf{C}_s = \{\mathbf{R}_1^s, \mathbf{R}_2^s, \ldots, \mathbf{R}_{|\mathcal{Y}_s|}^s\}$ of the seen category set as the prototypical knowledge. To assess the association between the sample and its category, we employ the semantic similarity to determine the score using the formula as $o_i = \hat{\mathbf{U}}^T \hat{\mathbf{R}}_i$ where $o_i$ is the score between the sample and the $i$-th category in $\mathbf{C}_s$. And the textual entity representation and entity-aware one are projected into the shared semantic space as $\hat{\mathbf{U}} = FFNN(\tilde{\mathbf{U}}; \theta_{o1}) \in \mathbb{R}^h$ and $\hat{\mathbf{R}}_i = FFNN(\tilde{\mathbf{R}}_i; \theta_{o2}) \in \mathbb{R}^h$. Based on the above score, we leverage the ranking loss to ensure

that the score of the true label remains higher than those of the false labels. The objective of max-margin ranking is defined as follows:

$$\mathcal{L}_{rank} = \sum_{i=1, y_i^s \neq Y}^{|Y_s|} \max(0, 1 - o^+ + o_i) \tag{13}$$

where $o^+$ is the score of the true type $y_i^s = Y$. And the objective function of Eqn. 7 can also be optimized by the task-relevant loss.

Given the batches of multimodal samples, we feed them into the model and firstly update the perturbation vector by Eqn. 12. To train the model with different objectives at once, we introduce the hyper-parameter to sum the corresponding losses. The overall loss is defined as follows:

$$\mathcal{L} = \mathcal{L}_{rank} + \beta \cdot (\mathcal{L}_{aux} + \mathcal{L}_{reg} + \mathcal{L}_{cl} + \mathcal{L}_{vat}) \tag{14}$$

where $\beta$ is the hyper-parameter to balance the different losses. Subsequently, we employ stochastic gradient descent (SGD) techniques to update model weights based on the loss calculated by Eqn. 14. In the inference phase, we evaluate the scores between samples in $D_{test}$ and unseen categories in $Y_u$, and designate the category scoring the highest as the predicted outcome.

## 5 Experiments

### 5.1 Datasets and Experiment Settings

We delve into the realm of zero-shot multimodal information extraction, specifically focusing on two tasks: multimodal named entity typing (MET) and multimodal relation extraction, both executed within a zero-shot setting. For these tasks, we undertake experiments utilizing the respective benchmark datasets. For MET task, we utilize the WikiDiverse [21] as the benchmark dataset. Each sample in the dataset consists of a text-image pair sourced from Wikinews, with the entity mention in the sentence manually annotated into one of 13 fine-grained types. For the MRE task, we utilize the benchmark dataset proposed by Zheng et al. [27], which is based on Twitter posts. Annotators randomly selected samples covering various topics. The MRE dataset contains samples categorized into 23 relation types. Since the above two datasets include meaningless categories such as "Other" or "None", we exclude them and only consider categories with actual semantics.

To compare our model with the baselines under the zero-shot setting, we mimic this scenario by randomly splitting original category set into three parts. For the MET task, we allocate 4 categories to each of the training, validation, and test sets individually. In the case of the MRE task, we assign 8, 7, and 7 categories to the training, validation, and test sets respectively. Ultimately, we conduct the experiments 3 times using different seeds and report the mean and standard deviation of the performances. The hidden layers are configured with a size of 768, while the hyper-parameter $\beta$, expert numbers $K$, learning rate and batch size are set to 1.0, 8, 1e-5 and 16, respectively. For both tasks, we set the training epochs to 20. All experiments are expedited using NVIDIA GTX A6000 devices.

### 5.2 Compared Methods

We compare MG-VMoE with both previous text-based ZS-IE models and multimodal models to demonstrate its effectiveness. For ZS-IE, the label-embedding-based prototype (Proto) network [11]

**Table 1: Performance comparison on the MET and MRE datasets under the zero-shot settings. The bold numbers indicate that the improvement of MG-VMoE over traditional baselines is statistically significant with $p < 0.01$ under t-test.**

| Multimodal Named Entity Typing | | | |
|---|---|---|---|
| Model | Precision | Recall | F1 | Accuracy |
| Text | | | | |
| Proto | 23.9 ± 5.9 | 24.6 ± 3.5 | 13.9 ± 5.7 | 29.3 ± 11.7 |
| DBZFET | 30.9 ± 4.9 | 29.5 ± 4.2 | 16.2 ± 1.3 | 23.6 ± 1.4 |
| NZFET | 26.3 ± 9.0 | 26.8 ± 4.8 | 13.3 ± 1.8 | 18.4 ± 2.6 |
| MZET | 29.9 ± 3.2 | 28.8 ± 5.6 | 11.4 ± 1.5 | 21.1 ± 6.5 |
| Multimodal | | | | |
| MMProto | 27.3 ± 3.8 | 27.2 ± 4.1 | 17.1 ± 9.2 | 24.4 ± 12.4 |
| MOVCNet | 28.6 ± 5.0 | 24.0 ± 3.3 | 13.9 ± 5.0 | 23.3 ± 10.8 |
| LLaVA | 40.9 ± 2.8 | 51.6 ± 6.4 | 39.5 ± 2.9 | 53.5 ± 9.7 |
| Ours | **37.1** ± 3.3 | **31.3** ± 4.9 | **23.7** ± 2.0 | **45.1** ± 2.0 |

| Multimodal Relation Extraction | | | |
|---|---|---|---|
| Model | Precision | Recall | F1 | Accuracy |
| Text | | | | |
| Proto | 24.5 ± 12.5 | 27.2 ± 10.2 | 21.6 ± 9.6 | 38.0 ± 9.3 |
| ZS-BERT | 29.5 ± 4.8 | 29.3 ± 5.4 | 22.9 ± 4.1 | 37.2 ± 8.7 |
| RE-Matching | 28.0 ± 4.8 | 31.1 ± 7.1 | 22.7 ± 11.1 | 32.4 ± 17.4 |
| ZS-SKA | 29.9 ± 7.4 | 21.8 ± 6.0 | 16.6 ± 7.1 | 26.8 ± 11.3 |
| Multimodal | | | | |
| MMProto | 33.8 ± 3.9 | 29.8 ± 4.5 | 23.4 ± 4.8 | 35.6 ± 6.2 |
| MOVCNet | 27.6 ± 11.1 | 25.4 ± 7.4 | 19.8 ± 6.2 | 33.8 ± 10.5 |
| LLaVA | 25.1 ± 4.6 | 18.2 ± 2.0 | 13.6 ± 0.7 | 14.9 ± 3.1 |
| Ours | **34.5** ± 8.1 | **32.8** ± 5.1 | **27.3** ± 8.1 | **40.0** ± 5.6 |

is a classic and effective baseline to perform the tasks. Besides, we select the description-based ZS-ET models including: DBZFET [16] and NZFET [17] as baselines which exploited the attention mechanisms to extract local features. And the memory augmented model MZET [23] was introduced for transferring knowledge from observed types to unobserved ones. For ZS-RE, Chen and Li [2] proposed ZS-BERT to encode entity and context representations and connect them with semantic representations of relations. Zhao et al. [25] exploited the fine-grained matching mechanism to extract the effective entity and context features. And Gong and Eldardiry [7] made use of prompt learning to learn the representations of both seen and unseen relations. For ZS-MIE, we extend Proto with the pre-trained vision model as the multimodal prototype network (MMProto) [20]. Zhang et al. [24] proposed the multimodal entity typing method (MOVCNet) to capture the semantic correlation between textual and visual representations. With the development of multimodal large language models (MLLM), we also select 13B version of LLaVA [10] as the baseline to perform the ZS-MIE tasks.

### 5.3 Experimental Results

We evaluated MG-VMoE alongside baseline models on the MET and MRE benchmark datasets, and reported macro-averaged precision (P), recall (R), F1 scores, and accuracy, considering the varying sample sizes across different categories. The detailed experimental results are shown in Table 1. Our model performed exceptionally well in most metrics on the MET dataset, achieving F1 and accuracy scores that exceeded the traditional baselines by 6.6% and 15.8%, respectively. Among text-based methods, DBZFET consistently

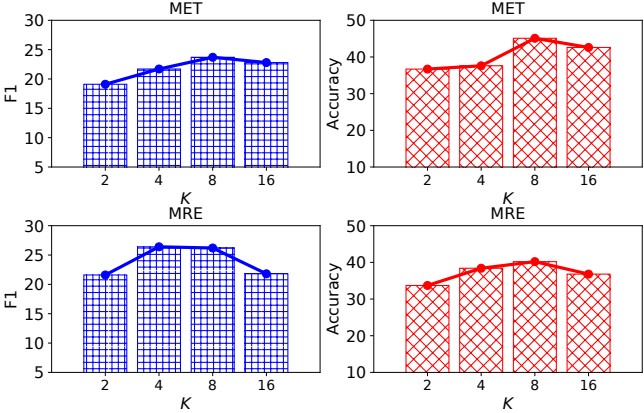

Figure 3: The performance comparison of MG-VMoE with different expert module numbers.

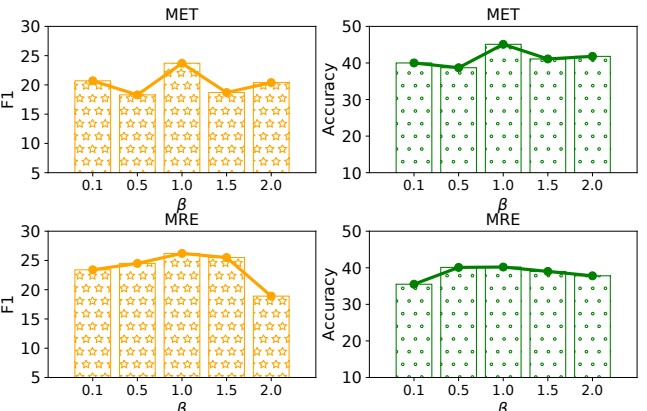

Figure 4: The performance comparison of MG-VMoE with different hyper-parameter $\beta$ values.

Table 2: The ablation study results of MG-VMoE on the MET and MRE benchmark datasets.

| Model | MET | | MRE | |
|---|---|---|---|---|
| | F1 | Accuracy | F1 | Accuracy |
| MG-VMoE | 23.7 | 45.1 | 27.3 | 40.0 |
| w/o VMoE | 19.4 | 29.2 | 20.8 | 36.0 |
| w/o MG-VAT | 18.2 | 38.9 | 21.5 | 32.5 |

outperformed the Proto model in terms of F1 score, emphasizing the significance of the attention module in capturing local sentence representations pertinent to entity types. We introduce the MG-VMoE model for fusing multimodal data to recognize novel types, consistently surpassing text-based models in performance, thereby demonstrating the superiority of our proposed approach. But LLaVA achieved better results than our model and we analyze that LLaVA is based on LLaMA [19] which was pre-trained on the web contents of Wikipedia. The MET dataset, sourced from the Wikinews website, includes entities that are also documented on Wikipedia. Therefore, the results of LLaVA are higher than those of the MG-VMoE model.

On the MRE dataset, our model surpassed all baseline models, achieving an F1 score that was 3.9% higher and an accuracy score that was 2.0% higher compared to the baselines. Furthermore, the multimodal-based MMProto exceeded text-based models in F1 score, highlighting the beneficial impact of visual information on ZS-MIE tasks. And LLaVA fails to surpass the proposed model, as it was not pre-trained on the Twitter content comprising the MET dataset. In conclusion, our model outperforms both text-based and multimodal-based baselines because of our novel fusion of text and image information using the VMoE network, combined with the creation of MG-VAT, which enhances multimodal representations and ultimately benefits MG-VMoE.

## 5.4 Ablation Study

The results demonstrate the significant role each component plays in determining the model's overall performance. To fully assess the effectiveness of the different modules introduced in MG-VMoE, an ablation study was performed, and its results are displayed in Table 2. Significantly, the exclusion of the variational mixture of experts (VMoE) caused a substantial decline in performance, underscoring the importance of integrating aligned and informative multimodal representations. This decline occurs because the low-level features from both modalities possess complementary information, which collectively enhances the comprehension of the input data. By utilizing VMoE to fuse these features, the model is able to capture fine-grained semantic correlations between the modalities, resulting in enhanced performance. Additionally, we assess the impact of multimodal graph-based virtual adversarial training (MG-VAT) on the ultimate outcomes. Our findings indicate that MG-VAT plays a

crucial role in boosting the model's final performance. By adopting an adversarial training strategy to refine multimodal representations at a fine-grained level, MG-VAT augments the model's capacity to precisely capture semantic connections among multimodal samples, leveraging graph-based information. To summarize, the results obtained confirm that employing the VMoE network architecture alongside the MG-VAT strategy for modeling detailed multimodal representations can improve model performance.

## 5.5 Influence of Experts Numbers

The VMoE network consists of expert modules, and their quantity plays a crucial role in the network's performance. To assess the impact of the number of experts, we conducted experiments to compare the performance of MG-VMoE with varying numbers of expert modules, as illustrated in Figure 3. The varying number of expert modules affects the MG-VMoE's performance on the two benchmark datasets, with optimal results achieved when the model includes 8 experts. A significant drop in performance occurs when reducing the number of experts in the model, as the experts are individually trained to capture multimodal representations of samples across various categories, and with fewer experts, effective learning of these representations becomes impractical. Increasing the number of experts in the model does not yield better results, as redundant experts may lead to overfitting on the training samples. Furthermore, the varying outcomes of MG-VMoE with different numbers of experts demonstrate the VMoE network's effectiveness.

## 5.6 Influence of Hyper-parameter $\beta$

During the training process, we uniformly train the model by summing the loss functions of VMoE and MG-VAT with the ranking loss, weighted by a hyper-parameter. To assess the impact of the

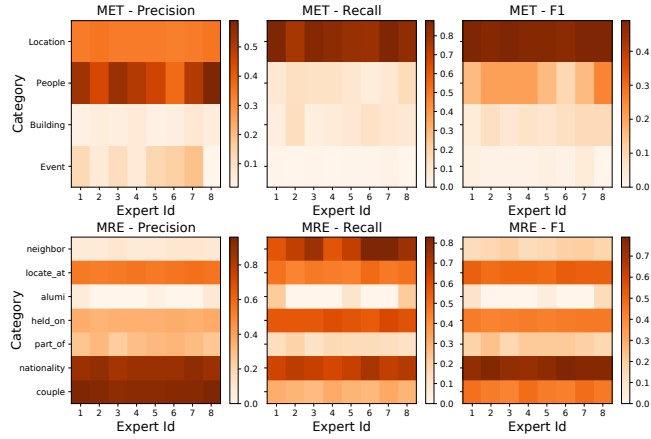

**Figure 5: The performance comparison of each expert module individually activated in MG-VMoE.**

hyper-parameter $\beta$, we perform parameter sensitive experiments as shown in Figure 4. The MG-VMoE's performance on the two benchmark datasets is influenced by various $\beta$ values. As $\beta$ value increases, the model focuses more on VMoE and MG-VAT losses, resulting in a significant drop in F1 scores. This is because the model fails to obtain a meaningful signal from the ranking loss and is unable to learn generalized representations for diverse categories. When $\beta$ value is reduced, the model struggles to obtain sufficient signals from VMoE and MG-VAT losses, leading to a decline in both F1 and accuracy scores. This is due to the model's inability to effectively leverage VMoE and MG-VAT, resulting in its failure to learn aligned and informative multimodal representations for different categories. Therefore, we achieve optimal results by balancing the ranking loss with other losses, setting the value of $\beta$ to 1.0.

### 5.7 Experts Abilities Analysis

During our experiments, MG-VMoE comprises 8 expert modules that are activated using sparse weights for forward propagation. To evaluate the capabilities of various experts, we examine the performance of each expert module independently activated within MG-VMoE, as illustrated in Figure 5. While the experts exhibit comparable overall performance, there are slight discrepancies in their prediction outcomes, particularly in terms of precision, recall, and F1 scores. Each expert acquires diverse multimodal representations for samples belonging to different categories. On the MET dataset, Expert #7 outperforms other experts in terms of results for the "Event" type, but falls behind Expert #8 for the "People" type. Furthermore, the recall scores of experts on the MRE dataset vary significantly. Considering the diverse capabilities of experts demonstrated in Figure 5, as well as the inferior performance of the model with fewer experts shown in Figure 3, having an optimal number of experts is essential for accurately modeling multimodal representations and enhancing the performance of MG-VMoE.

### 5.8 Visualization Analysis

In order to evaluate the effectiveness of multimodal representations for ZS-MIE, we visualize the features learned from MG-VMoE and baseline models in Figure 6. We choose samples from specific categories within the test sets and utilize both MG-VMoE and baseline

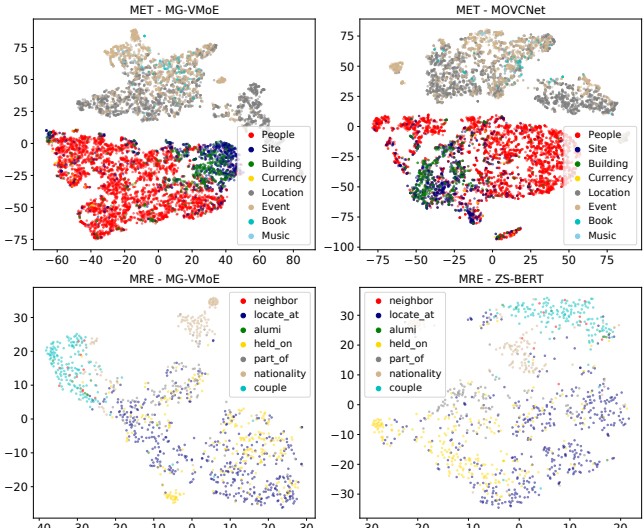

**Figure 6: Visualization of the t-SNE results exhibiting the multimodal representations of samples with particular categories, which were extracted individually from MG-VMoE and baseline models.**

models to obtain their respective representations. Subsequently, we employ t-SNE [12] to reduce the dimensionality of these output representations to two dimensions. The results show that the representations of MG-VMoE are more tightly grouped within each category, suggesting that MG-VMoE is more adept at distinguishing subtle variations among samples within the same category. For instance, the representations generated by MG-VMoE are more clustered compared to those produced by ZS-BERT on the MRE dataset. Furthermore, on the MET dataset, the representations of various categories from MG-VMoE exhibit a tighter grouping within each category. This can be explained by the abundant semantic content in multimodal data and the effectiveness of MG-VAT in capturing semantic relationships between samples. By integrating multimodal information using MG-VAT, the model acquires robust and transferable features, leading to improved performance in ZS-MIE tasks.

## 6 Conclusion

This paper investigates zero-shot multimodal information extraction (ZS-MIE) tasks, and mainly aims to address the coarse-grained multimodal representation learning limitation. To overcome this limitation, we introduce the multimodal graph-based variational mixture of experts (MG-VMoE) network tailored for ZS-MIE tasks. The MG-VMoE network builds upon fine-grained multimodal representation learning, incorporating both the variational mixture of experts (VMoE) and multimodal graph-based virtual adversarial training. Serving as the core, the VMoE network utilizes sparse weights to activate expert modules, where each expert functions as a variational information bottleneck (VIB) for extracting informative and aligned textual and visual representations. Meanwhile, the multimodal graph-based virtual adversarial training is employed to capture semantic correlations between multimodal samples and enhance the clustering tightness of samples within the same category. Experimental results demonstrate the generalization ability of MG-VMoE compared to baseline methods on ZS-MIE tasks.

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
