# OpenReview forum: "Multimodal Graph-based Variational Mixture of Experts Network for Zero-shot Multimodal Information Extraction"
_ACM.org/TheWebConf/2025/Conference — WWW 2025 Poster_

### Official Review · Reviewer_3JCZ · 2024-11-18

**Novelty:** 3
**Technical Quality:** 4

**Review:**

Overall Evaluation:
This paper presents a novel approach to zero-shot multimodal information extraction (ZS-MIE) by introducing the multimodal graph-based variational mixture of experts (MG-VMoE) network. While the technical implementation and experimental validation are somewhat solid, there are some issues:
1.Writing and Presentation: The manuscript requires improvement in terms of English language usage and clarity. The writing shows inconsistencies in grammar, article usage, and technical terminology. A thorough review by a native English speaker would help improve the overall presentation and readability of the paper.Also, there are some typo: e.g., "Other" or "None" in line 619
2.Figure Quality: The visual presentation of Figure 2 in this paper needs significant improvement in terms of professional appearance, consistency, and clarity. Make consistent for all characters in this figure.
3.Experimental Validation: The experimental validation could be enhanced by expanding to additional datasets, including more recent baseline comparisons, and adding statistical significance tests. The analysis should explore more challenging scenarios and edge cases to fully demonstrate the model's capabilities and limitations. In Table 1 for “Ours”, why you only list the improvement without showing the exact accuracy. The concept is not clear, is the improvement the absolute value, and who takes as the baseline?
4.Practical Considerations: The paper would benefit from a more thorough discussion of practical aspects, including computational efficiency, resource requirements, and real-world deployment challenges. Due to high cost may appear in Graph construction, how can your approach efficiently applied in the large-scale web scenario? Also, Case studies demonstrating practical applications and guidelines for parameter selection in different scenarios would make the work more applicable to real-world situations.
5.Related work: No discussion about Variational Mixture of Experts Network
6.Please highlight the motivation of the paper
7.As shown in Eq.(14), L_aux, L_reg, L_cl, and L_vat exhibit identical weight, why this setting is reasonable.
Summary: This paper presents a valuable contribution to the field of multimodal information extraction, particularly in zero-shot settings. The proposed MG-VMoE network demonstrates impressive performance and introduces several innovative technical components. While the paper would benefit from improved English writing, broader experimental validation, and enhanced figure quality, the current work represents a significant step forward in multimodal learning. The research is well-executed and the results are convincing, making it worthy of publication after addressing the suggested improvements.

**Questions:**

Overall Evaluation:
This paper presents a novel approach to zero-shot multimodal information extraction (ZS-MIE) by introducing the multimodal graph-based variational mixture of experts (MG-VMoE) network. While the technical implementation and experimental validation are somewhat solid, there are some issues:
1.Writing and Presentation: The manuscript requires improvement in terms of English language usage and clarity. The writing shows inconsistencies in grammar, article usage, and technical terminology. A thorough review by a native English speaker would help improve the overall presentation and readability of the paper.Also, there are some typo: e.g., "Other" or "None" in line 619
2.Figure Quality: The visual presentation of Figure 2 in this paper needs significant improvement in terms of professional appearance, consistency, and clarity. Make consistent for all characters in this figure.
3.Experimental Validation: The experimental validation could be enhanced by expanding to additional datasets, including more recent baseline comparisons, and adding statistical significance tests. The analysis should explore more challenging scenarios and edge cases to fully demonstrate the model's capabilities and limitations. In Table 1 for “Ours”, why you only list the improvement without showing the exact accuracy. The concept is not clear, is the improvement the absolute value, and who takes as the baseline?
4.Practical Considerations: The paper would benefit from a more thorough discussion of practical aspects, including computational efficiency, resource requirements, and real-world deployment challenges. Due to high cost may appear in Graph construction, how can your approach efficiently applied in the large-scale web scenario? Also, Case studies demonstrating practical applications and guidelines for parameter selection in different scenarios would make the work more applicable to real-world situations.
5.Related work: No discussion about Variational Mixture of Experts Network
6.Please highlight the motivation of the paper
7.As shown in Eq.(14), L_aux, L_reg, L_cl, and L_vat exhibit identical weight, why this setting is reasonable.
Summary: This paper presents a valuable contribution to the field of multimodal information extraction, particularly in zero-shot settings. The proposed MG-VMoE network demonstrates impressive performance and introduces several innovative technical components. While the paper would benefit from improved English writing, broader experimental validation, and enhanced figure quality, the current work represents a significant step forward in multimodal learning. The research is well-executed and the results are convincing, making it worthy of publication after addressing the suggested improvements.

**Reviewer Confidence:**

4: The reviewer is certain that the evaluation is correct and very familiar with the relevant literature

**Scope:**

3: The work is somewhat relevant to the Web and to the track, and is of narrow interest to a sub-community

---

### Official Review · Reviewer_MpjG · 2024-11-27

**Novelty:** 5
**Technical Quality:** 6

**Review:**

This paper aims to solve two tasks in multimodal information extraction: multimodal named entity typing and multimodal relationship extraction. And it primarily focuses on the zero-shot multimodal information extraction.
1. This paper proposes two modules, VMoE and MG-VAT, to solve the semantic gap between the textual and visual modalities (local semantic relationships between image and text; semantic nuances between multimodal samples).
2. The experimental work of this paper is relatively sufficient, including ablation experiment and discussion of hyper-parameters. At the same time, there are some visualized work for quantitative analysis to support the paper.

**Questions:**

1. The statement in section 4.1 seems incomplete. ('Formally, The extended sentence with the special tokens is denied as. ')
2. The solid line and dotted line in Figure 1 are not properly marked and explained.
3. The formal definition of the task is ambiguous.
3. This paper lacks case study for explanation to highlight the effectiveness of each module. On the one hand, it can enhance the persuasiveness of this paper, and on the other hand, it can also make the description of the task more vivid, rather than just a formal definition.
4. Will all the source code of the author be released in the future?
5. Has the author conducted experiments on other multimodal large language models? Can you provide this information?

**Reviewer Confidence:**

3: The reviewer is confident but not certain that the evaluation is correct

**Scope:**

3: The work is somewhat relevant to the Web and to the track, and is of narrow interest to a sub-community

---

### Official Review · Reviewer_62gL · 2024-11-30

**Novelty:** 5
**Technical Quality:** 4

**Review:**

The paper proposes a new Zero-Shot Multimodal Information Extraction (ZS-MIE) method that combines Variational Mixture of Experts (VMoE) and Virtual Adversarial Training (MG-VAT) to optimize multimodal Named Entity Recognition (MET) and Multimodal Relation Extraction (MRE) tasks. The method presented in the paper demonstrates superiority in multimodal feature fusion and model training strategies, and shows promising results in experiments. The paper provides rich experimental data, including comparisons with existing baseline models, ablation studies, and hyperparameter sensitivity analysis, which effectively validate the effectiveness of the proposed method.

Advantages:

1. Novel Method Design: The strategy combining VMoE and MG-VAT proposed in the paper is innovative for multimodal information extraction tasks. This method effectively aligns and fuses multimodal semantic representations, particularly demonstrating strong generalization ability in a zero-shot setting.

2. Thorough Experimental Validation: The paper validates the model’s effectiveness and robustness through experiments on multiple benchmark datasets. Additionally, in-depth analyses such as ablation studies and hyperparameter sensitivity analysis provide strong support for the superiority of the model.

3. Significant Domain Impact: Zero-shot multimodal information extraction is a major challenge in current research, especially for tasks like Named Entity Recognition (MET) and Relation Extraction (MRE), which have high research and application value.


Disadvantages:

1. Insufficient Theoretical Analysis: Although the model design proposed in the paper is innovative, it lacks sufficient theoretical analysis, particularly regarding how VMoE and MG-VAT work together and interact. The paper focuses more on presenting experimental results and less on in-depth exploration of the principles behind the model design and its theoretical foundation, especially how these two modules work together to optimize the multimodal information extraction task.

2. Lack of Analysis on the Synergistic Effects and Interdependencies Between Modules: While the paper mentions the role of VMoE and MG-VAT in multimodal representation learning, there is insufficient analysis of how these two modules work together to enhance model performance. The specific interaction mechanisms between the two modules, how they cooperate during training to improve overall performance, especially whether there is any feedback mechanism, and how their synergy affects the final model performance, are not fully explored.

3. Absence of Ablation Study on Module Interactions: Although the paper conducts ablation studies, it does not explicitly compare the effects of using VMoE and MG-VAT individually with their combined use. Further ablation research, particularly comparing the performance of the two modules when used alone and in combination, could more clearly demonstrate their complementarity and their contribution to the overall model’s performance. This would help verify whether their effective integration indeed leads to significant performance improvements.

**Questions:**

Q1: The paper mentions that the two modules, VMoE and MG-VAT, play a key role in the multimodal information extraction task but does not explain in detail how these two modules collaborate during the training process. Could you further elaborate on how these modules collaborate to optimize multimodal representations during training? Are there any theoretical foundations or experimental results that support the mechanism of their cooperation?

Q2: The paper lacks a specific theoretical derivation of the interaction between VMoE and MG-VAT in the model. In particular, how does MG-VAT enhance the performance of VMoE through adversarial training? What is the specific enhancement mechanism? Is there a feedback mechanism or other forms of interaction between these two modules, and can this be verified through experiments or analysis?

Q3: Although the paper includes ablation experiments, it does not provide a detailed comparison of the performance differences when using VMoE and MG-VAT individually versus when they are combined. Could you further verify the specific contribution of the joint use of VMoE and MG-VAT to model performance through additional ablation or comparative experiments? If possible, could you show a specific performance comparison between using these modules individually and together?

Q4: The paper mentions that MG-VAT enhances the multimodal representation capabilities, but does not go into detail about its specific mechanism. Could you further explain how MG-VAT improves the robustness of multimodal representations through adversarial training, especially in a zero-shot learning setting, and how it helps the model handle unseen categories?

Q5: The experimental results section in the paper is relatively focused, and there is a lack of in-depth discussion on the theoretical design of the model. Could you provide a more detailed theoretical analysis, explaining why the Variational Mixture of Experts (VMoE) and Virtual Adversarial Training (MG-VAT) modules were chosen, and clarify their specific contributions and significance to the model’s performance?

**Reviewer Confidence:**

3: The reviewer is confident but not certain that the evaluation is correct

**Scope:**

4: The work is relevant to the Web and to the track, and is of broad interest to the community

---

### Official Review · Reviewer_29m8 · 2024-11-30

**Novelty:** 6
**Technical Quality:** 6

**Review:**

To solve the lack of fine-grained semantic correlation for existing zero-shot information extraction methods in Multimodal models, this paper introduce the a novel MG-VMoE, which based on on the MoE framework and leverages sparse expert weights for precise alignment of multimodal representations.

Pros:
(1). This paper demonstrates a clear and well-organized structure, ensuring that the flow of information is logical and easy to follow

(2). This paper introduces notable novelty by presenting innovative contributions to the field

(3). This paper features comprehensive experiments and an in-depth ablation study, underscoring the robustness and validity of its findings.

Overall, this is a commendable and publish-worthy paper that makes meaningful contributions to its field.

**Questions:**

(1). How does the computational cost of this method compare to other existing approaches? Is its inference speed acceptable compared to LLaVA?

(2). In Table 1, the best-performing LLaVA should be bolded to avoid confusion.

**Reviewer Confidence:**

2: The reviewer is willing to defend the evaluation, but it is likely that the reviewer did not understand parts of the paper

**Scope:**

4: The work is relevant to the Web and to the track, and is of broad interest to the community

---

### Official Review · Reviewer_4G7C · 2024-12-03

**Novelty:** 4
**Technical Quality:** 3

**Review:**

Summary:

This paper introduces a multi-modal graph-based variation mixture of experts network for zero-shot multimodal information extraction problem. The motivation of the work is current methods of zero-shot IE only considered text ignore the multimodal information, and the multimodal IE methods ignore the fine-grained semantic correlation of the text-image pairs and samples. Thus authors propose the MG-VMoE methods to model the fine-grained semantic correlations and is capable of achieve MIE in the zero-shot setting.

Strengths:

1. This paper focuses on the zero-shot multimodal information extraction task, which seems hasn’t been fully explored and analyzed in the previous works.
2. The MC-VMoE method proposed in this paper achieve good results on MIE tasks in the zero-shot setting.
3. The analysis of the MC-VMoE model helps readers to understand how the model works and which module is the most important.

Weaknesses:

1.  The main motivation for the model design is to capture the fine-grained semantic correlation between text-image pairs. The motivation of capturing fine-grained semantic correlations is a common challenge for MIE methods and is not closely associated with zero-shot MIE. But the experiments are conducted in the zero-shot setting rather than normal MIE setting. Thus there is a mismatching between the motivation, method design, and experiment setting.
2. The textual entity representation/ entity-aware representation and semantic representation of categories are got through different modules with no interactions. The proposed method could solve the zero-shot MIE problem is mainly due to the BERT could encode any labels into the same representation space. Thus the technique contribution of the work is limited in terms of solving zero-shot problems.
3. The experiments are conducted on zero-shot MIE setting. While the baseline methods are either proposed in the normal MIE setting, or the zero-shot IE setting with only text considered. None of the them are proposed or adapted to zero-shot MIE. At least one or two methods that are designed to handle zero-shot MIE problem are expected to be compared, to help readers judge how efficient the MG-VMoE method is from the perspective of ZS MIE.

**Questions:**

1. What is the difference between the multimodal representation space f Traditional MIE model and MG-VMoE model? What is difference between the dash line and the solid line in the multimodal representation space figure?
2. Could you explain semantic variation of sample within the same category with concrete examples?
3. The best and the worst performed category of different experts in Figure 5 almost are the same. The discrepancies between different experts are small. How should we interpret this results and prove the effectiveness of MoE at the same time?

**Reviewer Confidence:**

3: The reviewer is confident but not certain that the evaluation is correct

**Scope:**

3: The work is somewhat relevant to the Web and to the track, and is of narrow interest to a sub-community